# Up-down biphasic volume response of human red blood cells to PIEZO1 activation during capillary transits

**Simon Rogers** [1] *, **Virgilio L. Lew** [2] *

1 School of Computing Science, University of Glasgow, United Kingdom, 2 Physiological Laboratory, Department of Physiology, Development and Neuroscience, University of Cambridge, Downing Site, Cambridge, United Kingdom

* simon.rogers@glasgow.ac.uk (SR); vll1@cam.ac.uk (VLL)

**Data Availability Statement:** All relevant data are within the manuscript and its Supporting Information files.

**Funding:** The authors received no specific funding for this work.

## Abstract

In this paper we apply a novel JAVA version of a model on the homeostasis of human red blood cells (RBCs) to investigate the changes RBCs experience during single capillary transits. In the companion paper we apply a model extension to investigate the changes in RBC homeostasis over the approximately 200000 capillary transits during the ~120 days lifespan of the cells. These are topics inaccessible to direct experimentation but rendered mature for a computational modelling approach by the large body of recent and early experimental results which robustly constrain the range of parameter values and model outcomes, offering a unique opportunity for an in depth study of the mechanisms involved. Capillary transit times vary between 0.5 and 1.5s during which the red blood cells squeeze and deform in the capillary stream transiently opening stress-gated PIEZO1 channels allowing ion gradient dissipation and creating minuscule quantal changes in RBC ion contents and volume. Widely accepted views, based on the effects of experimental shear stress on human RBCs, suggested that quantal changes generated during capillary transits add up over time to develop the documented changes in RBC density and composition during their long circulatory lifespan, the quantal hypothesis. Applying the new red cell model (RCM) we investigated here the changes in homeostatic variables that may be expected during single capillary transits resulting from transient PIEZO1 channel activation. The predicted quantal volume changes were infinitesimal in magnitude, biphasic in nature, and essentially irreversible within inter-transit periods. A sub-second transient PIEZO1 activation triggered a sharp swelling peak followed by a much slower recovery period towards lower-than-baseline volumes. The peak response was caused by net $CaCl_2$ and fluid gain via PIEZO1 channels driven by the steep electrochemical inward $Ca^{2+}$ gradient. The ensuing dehydration followed a complex time-course with sequential, but partially overlapping contributions by KCl loss via $Ca^{2+}$-activated Gardos channels, restorative $Ca^{2+}$ extrusion by the plasma membrane calcium pump, and chloride efflux by the Jacobs-Steward mechanism. The change in relative cell volume predicted for single capillary transits was around $10^{-5}$, an infinitesimal volume change incompatible with a functional role in capillary flow. The biphasic response predicted by the RCM appears to conform to the quantal hypothesis, but whether its cumulative effects could account for the documented changes in density during RBC senescence

**Competing interests:** The authors have declared that no competing interests exist.

required an investigation of the effects of myriad transits over the full four months circulatory lifespan of the cells, the subject of the next paper.

## Author summary

Each human red blood cell traverses narrow capillaries about 1000 to 2000 times per day. Cell deformability is essential for smooth transits and this requires tight control of cell volume well below spherical maxima. The pump-leak ion-flux balance controlling red cell volume involves sodium-potassium and calcium pumps and a constellation of passive membrane transporters. PIEZO1, a mechanosensitive ion channel plays a central role, activating transiently during capillary transits. The main question, not accessible to direct experimentation, is what happens to red cell volume and composition *in vivo* during each capillary transit. We attempted answering this question by applying novel extensions to a tried and tested model of red cell homeostasis that encodes the current knowledge on the subject. The results revealed an unexpected up-down biphasic volume response characterized by a sharp initial surge caused by $CaCl_2$ influx via PIEZO1, associated with osmotic fluid gain, followed by a slow reversal, a prediction in conflict with prevailing views. The infinitesimal amplitude of the predicted volume changes precludes them playing a significant role in flow dynamics. In the following paper we explore how the biphasic response helps explain the age-related changes red cells experience during their long circulatory lifespan.

## Introduction

We apply here a new multiplatform JAVA-based mathematical-computational model of the homeostasis of human red blood cells (RBCs) to investigate the changes RBCs experience during capillary transits and throughout their long circulatory lifespan in vivo. There is a vast literature on the circulatory behaviour and rheology of RBCs in health and disease, and on many of the age-related changes in metabolism, haemoglobin properties and membrane transport, all topics amenable to experimentation in vitro. The one aspect not accessible to experimentation concerns the dynamics of the homeostatic changes RBCs experience in the circulation both during single capillary transits and throughout their entire four months long lifespan. There are abundant timed snapshots of experimental data on age-related changes in RBC volume, density, pH, membrane potentials, ion composition and membrane transport, a solid body of homeostatic information, but no mechanistic understanding of the continuum of dynamic interactions shaping these changes, and no way of testing the feasibility of hypothetical suggestions.

The most important recent findings concern the expression and function of mechanosensitive PIEZO1 channels in the RBC membrane. Together with the large body of early data it has now become possible to constrain the parameter space sufficiently and attempt a computational modelling approach to unravel the mechanisms behind the homeostasis changes individual RBCs experience throughout the myriad capillary transits in the circulation. The results of this investigation, reported in this and the next paper [1], present an entirely new and unexpected perspective on the coordinated roles played by five RBC membrane transporters to enable the long circulatory lifespan of RBCs. We briefly review next relevant background information on the model and on red blood cells homeostasis.

## The red blood cell model (RCM)

The core of the new red cell model (RCM) is offered with open access from a GitHub repository together with a comprehensive user guide and tutorial (https://github.com/sdrogers/redcellmodeljava). The model operates as a RCM*.jar programme within the JAVA environment which needs to be preinstalled. The file name contains coded information on date and update status. Operation of the RCM model generates protocol files as editable *.txt files, and the results of simulations are reported in *.csv files. The columns of the *.csv files record the time-changes for each of the homeostatic variables of the modelled system composed of cells and suspending medium. The Appendix at the end of this paper provides a detailed account of the governing equations of the RCM. S1 Fig introduces the user interface for the Welcome and Central pages of the RCM. For all further details on the broad range of model applications in research, clinical investigation and teaching the reader is referred to the User Guide in the repository.

## A brief primer on the biology and homeostasis of human red blood cells

The main functions of RBCs, the transport of $O_2$ and $CO_2$ between lungs and tissues, evolved to operate with minimal energy cost to the organism. A first critical feature enabling such economy is the extremely low cation permeability of the RBC membrane [2–4]. This allows the cells to maintain steady volumes for extended periods of time with minimal cation traffic, pump-leak turnover rates and ATP consumption. RBCs are the most abundant cells in the body. Yet their glycolytic ATP turnover amounts to less than 0.06% of the total body ATP turnover in healthy human adults [5].

A second critical feature of the optimized economy concerns the compromise between RBC turnover rate and circulatory lifespan. RBC mass was adapted to provide for adequate gas transport at all levels of physiological demand. Biosynthetic and biodegradable replacement of such a large cell mass imposes a heavy metabolic cost to the organism which can only be reduced by extending the circulatory lifespan of the cells thereby reducing their replacement frequency. Circulatory longevity, on the other hand, is limited by the extent to which "tight and cheap" RBCs, devoid of nucleus, organelles, and biosynthetic capacity, and relying only on glycolytic metabolism, can sustain the functional competence required for volume stability and optimal rheology. Optimal rheology requires that the RBCs retain a large degree of deformability for rapid passage through narrow capillaries and for ensuring minimal diffusional distances for gas-exchange across capillary walls. Deformability, in turn, depends on the RBCs maintaining their volume well below their maximal spherical volume (reduced volume), a condition fulfilled when their reduced volume is kept within a narrow margin around 60% [6,7]. For human RBCs with a mean circulatory lifespan of about 120 days, this represents a substantial challenge, and the mechanism by which this is achieved is the subject of the investigation in this and the next paper [1].

RBCs undergo gradual densification for most of their lifespan [8,9]. After about 70–90 days, the densification trend reverses [10–13], a phenomenon interpreted as a strategy evolved to prolong the circulatory survival of the cells by keeping them within the optimal reduced volume range, otherwise imperilled by sustained dehydration. The terminal clock mechanism for RBC removal from the circulation is an active field of research with important recent contributions [14–19], but not addressed in the investigation reported here. There is solid evidence documenting age-dependent late reductions in the activity of the Na/K pump (ATP1A1 gene) [20–22], gradual decline in $Ca^{2+}$ pump activity (PMCA, PMCA4b, ATP2B4 gene) [23–26], and in Gardos channel activity (KCNN4) [27] among membrane transporters, and in the activities of other enzymes and membrane components [8], reductions generally attributed to unavoidable decay and damage in a cell deprived of protein renewal capacity.

## PIEZO1 in RBCs

Pioneering findings in the eighties suggested that shear stress in the circulation triggered the progressive dehydration of RBCs via $Ca^{2+}$-mediated activation of Gardos channels [28,29], suggesting that $Ca^{2+}$ influx was mediated by activation of mechanotransduction pathways in the RBC membrane. Progress in this area was advanced only recently with independent discoveries reported almost simultaneously in PLoS ONE early in 2010 [30,31], before the actual expression of PIEZO1 in the RBC membrane had been firmly confirmed. Vandorpe et al., [30] demonstrated that Psickle, the permeability pathway activated by deoxy-Hb S polymers in sickle RBCs of human and HbS-mice could be blocked by the PIEZO1 channel inhibitor GsMTx-4 providing strong support for the PIEZO1 identity of Psickle. Dyrda et al., [31] showed that local membrane deformations activate $Ca^{2+}$-dependent $K^+$ and anionic currents in intact human RBCs. Work in the following years firmly established the expression and activity of stress-activated PIEZO1 channels in human RBC membranes and the causal role of mutant PIEZO1 channels in certain haemolytic anaemias [7,32–37], setting PIEZO1 as the prime candidate pathway to account for the observed deformation-induced increase in $Ca^{2+}$ permeability [31,37].

## Methods

### Comparison between RCM- predicted and observed $Ca^{2+}$ signals during flow-induced RBC deformations in microfluidic chambers

Elegant experiments by Danielczok et al., [38] have recently shown that Fluo-4-loaded RBCs circulating through microfluidic chips elicited transient calcium signals only during passage through constricted domains in the microfluidic channels, signals blocked by the PIEZO1 channel inhibitor GsMTx. These results are particularly relevant here because they represent the closest experimental approximation to capillary flow conditions available to date. Their results strongly support a firm association between reversible RBC deformation, reversible PIEZO1 activation and reversible calcium influx during the deformation stage, offering a unique opportunity to test the model capacity to emulate the calcium signal sequence by comparison with firm experimental results, as attempted in Fig 1. It can be seen that the model can easily accommodate the pattern of the observed $Ca^{2+}$ responses elicited by PIEZO1.

Because PIEZO1 channels are poorly ion selective [39], each capillary passage would be expected to trigger gradient dissipation for $K^+$, $Na^+$, $Ca^{2+}$, $Mg^{2+}$ and $Cl^-$, creating minute changes in RBC ion contents. Because of its huge inward electrochemical gradient even a minor and brief increase in $Ca^{2+}$ permeability could generate a significant elevation in $[Ca^{2+}]_i$ above baseline levels, a signal with the power to activate $Ca^{2+}$-sensitive Gardos channels thus amplifying downstream effects of KCl loss and cell dehydration. The cumulative effects of such quantal steps during capillary transits were assumed to drive the gradual densification of RBCs during most of their circulatory lifespan, the quantal hypothesis [5,28].

We investigate here the changes in RBC variables predicted for single capillary transits applying the PIEZO routine of the RCM. At each stage in the course of this investigation compliance with reliable experimental data was used as the constraining guide on the range of acceptable model outcomes and parameter values.

### Modelling strategy

**Experimental derived constraints.**   From the start, it was critical to select the body of knowledge we wished the model outcomes to comply with, and to establish a hierarchical scale of weight for the different parameters based on experimental support and relevance. The

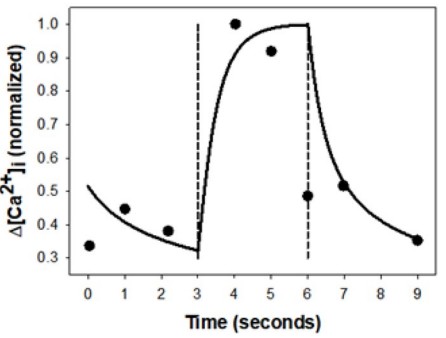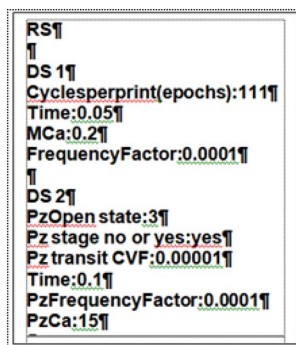

**Fig 1. Comparison between predicted (solid line) and measured (circles) patterns of intracellular $[Ca^{2+}]_i$ change induced by reversible PIEZO1 channel activation.** Measured and predicted $[Ca^{2+}]_i$ signals were normalized to maxima of 1 for comparison. The experimental results (circles) were redrawn from Fig 3Bb in Danielczok et al., [38]. Their Fig 3B shows fluorescent confocal images of a Fluo-4 loaded RBC before, during and after passing through a constriction in a microfluidic channel. In Fig 3Bb they show the corresponding changes in Fluo-4 $Ca^{2+}$ signals from which the points shown here were taken. The pattern reported for this figure was observed in 51 additional measurements as shown in their Fig 3C. The solid line representing the model prediction was obtained with the protocol shown next to the figure, simulating a 9s data collection period and a 3s passage through the constriction domain (indicted between the vertical lines). The experimental points suggest a maximal $Ca^{2+}$ signal by the 4th second, and near full recovery of the signal by the 6th second. At the low external calcium concentrations and extended open state (OS) conditions emulated in the protocol, calcium gain is much reduced and Gardos channels are minimally activated thus reducing the volume response to negligible levels. The real OS duration of the PIEZO1 channels within the 3s deformation period is unknown. The protocol applied for the solid curve simulation assumed a 3s OS duration and a PIEZO1 $Ca^{2+}$ permeability (PzCa) of 15 h⁻¹. However, alternative combinations of these two variables were found to produce similar fits of the data-set in Fig 3C as long as the product OS*PzCa was kept close to 45s/h in the simulations.

experimental data chosen to constrain and guide this investigation was primarily based on results by Vandorpe et al., [30], Dyrda et al., [31], Glogowska et al., [40], Danielczok et al., [38], Kuchel and Shishmarev, [37], Zhao et al., [39] Ganansambandam et al., [41], and also on dated results that harmonize with the more recent findings [42–44]. Five PIEZO1- attributed parameters defined the model protocol for single capillary transit simulations: duration of the open-state (OS, in s), three cationic permeabilities for $Ca^{2+}$, $Na^+$ and $K^+$, labelled PzCa, PzNa, PzK, respectively (in h⁻¹), and one anionic permeability for the combined diffusible $Cl^- + HCO_3^-$ anions, PzA (h⁻¹) (see S1 Governing Equations).

It is important to stress here that the attribution of an anion conductance to PIEZO1 (PzA) is only intended as a means to implement a minimalist phenomenology that enables us to investigate the need or otherwise to associate the changes in PIEZO1-mediated cation and anion permeabilities during the PIEZO1 open state in the simulations. As will become apparent in this and the following paper [1], the results do support a strong association between increased cation and anion permeabilities during the PIEZO1 open states. However, the model results carry no information on whether the anion permeability is through PIEZO1, as PzA may imply, or through another PxA pathway. The association may be hypothetically explained by interactions between different channels. For instance, the kinetics of PIEZO1 and that of the slippage conductance through the anion exchanger, AE1 [45,46], may be somehow linked through their connections to the underlying cytoskeletal mesh, generating a transient AE1 configuration with enhanced conductance during the PIEZO1 open state. Similar arguments may apply to other anionic conductance pathways described in the literature [31,40,47].

**The changing cell-medium proportions during capillary transits.** Images of RBCs during capillary transits show RBCs squeezed, elongated and deformed in the flow, often in single filing trains. During these brief transits, of between 0.5 and 2s duration, the volume of plasma surrounding the cells within the capillary becomes much smaller than the actual volume of the

cells, a condition that can be represented in the simulations with cell volume fractions (CVF) near one. During such squeezes any PIEZO1-mediated flux could substantially alter the composition of the medium surrounding the cells. During inter-transit periods, on the other hand, before RBCs ingress to, or after they emerge from capillaries, RBCs flow in the systemic circulation which acts like an infinite reservoir of a composition regulated by RBC-independent processes. Modelling such transit-inter-transit transitions requires a protocol which ensures that the composition changes within the miniscule volumes of medium surrounding the cells during the brief capillary transits are not carried over and accumulated in the medium surrounding the cells during inter-transit periods in the systemic circulation. This required introducing a "restore medium" subroutine for inter-transit periods to minimize carryover effects. Because transit times are so much shorter ($<$ 2s) than inter-transit periods (1–2 min) it may be questionable whether the brief high-CVF transitions could generate relevant effects, a question answerable only by comparing predicted outcomes with high and low CVF values during capillary transits. As we shall see, negligible effects at single capillary transits turned out to be of cumulative relevance over lifespans [1].

**The reference protocol ("Ref") (Fig 2).**   We carried out a large number of preliminary tests with the RCM, seeking compliance with documented experimental results. Four transport systems were found to be the main determinants of the response to deformation-induced changes in RBC homeostasis: PIEZO1 with PzA, the trigger of all downstream effects, Gardos channels, the PMCA and the Jacobs-Stewart mechanism (JS). The pattern that complied with known results could only be implemented with a rather restricted set of parameter values.

To report in the simplest possible terms the process followed to arrive at this result, and to explain the mechanisms involved, we generated a standardized Reference protocol (Fig 2) which delivers the curves labelled "Ref" in the figures (in black). The effects of parameter variations are shown by comparison to Ref. The Ref protocol runs a standard baseline period of two min. Ingress of a RBC in a capillary is simulated with an instant CVF transition from 0.00001 to 0.9 together with PIEZO1 activation. After 0.4s, egress is simulated with instant PIEZO1 closure and with the RBC returning to the systemic circulation with a CVF of 0.00001 and a medium composition restored to the original plasma-like condition. The unrealistically long duration of the post-transit periods used in the figures was intended to provide adequate information on the recovery rates to be expected for the different variables altered during the brief PIEZO1 open state. The default permeability values set for the Ref routine were: PzCa, 70h$^{-1}$; PzA, 50h$^{-1}$; and zero for PzNa and PzK. The reasons for these choices will become clear with the analysis of the results in this paper; a full explanation for the value of 70h$^{-1}$ attributed to the reference PzCa will have to await the analysis in the section on "Hyperdense collapse" in the companion paper [1], where it is shown that PzCa values compatible with observed lifespan densification patterns and cell integrity cannot exceed certain maximal values, as with the 70h$^{-1}$ value linked to a PIEZO1 open state duration of 0.4s in the reference protocol.

**Preliminary RCM tests.**   Points noted during preliminary tests, of secondary relevance but considered necessary to retain as background information, are succinctly summed up next. Interested readers are encouraged to design protocols and replicate these tests with the RCM. 1) It made little difference to the results whether RBCs were simulated emerging from the capillary immediately after PIEZO1 closure or after lingering in the high CVF conditions for up to 2s after PIEZO1 closure. This validated the use of simpler protocols with the same duration set for PIEZO1 open states and transit times (Fig 2). 2) For the effects of PIEZO1 on ion fluxes, open state duration within the time-scale of these simulations, proved inversely related to ion permeability. For instance, the calcium influx generated by a PzCa of 70h$^{-1}$ over 0.4s was similar to that generated by a PzCa of 140h$^{-1}$ over 0.2s. Thus, OS duration and permeabilities are not independent parameters in the context of their modelled effects. 3)

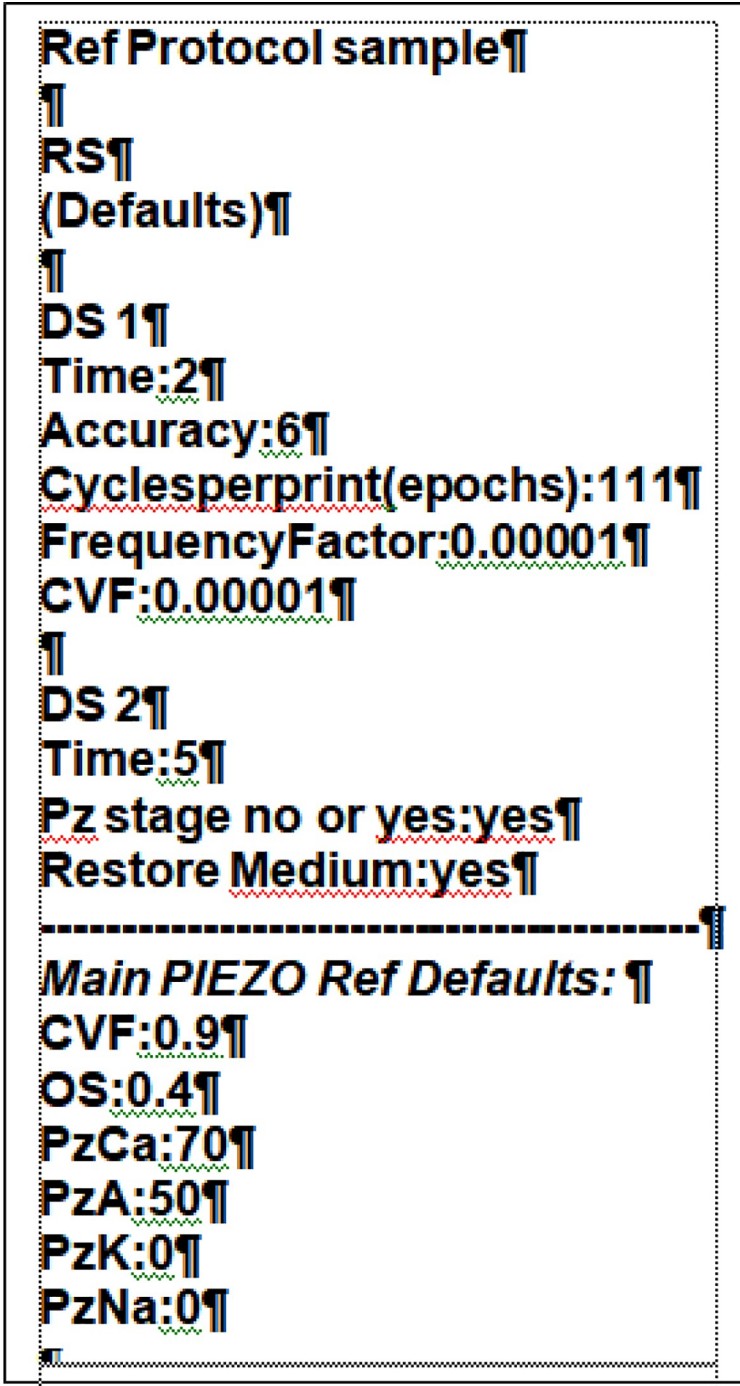

**Fig 2. Default Reference protocol (Ref) for testing the effects of parameter and variable changes on RBC responses during and after capillary transits.** The default protocol runs a three-step dynamic sequence on a RBC configured by the defaults in the Reference State (RS): DS1: a two minute baseline; DS2: a pre-programmed routine of two distinct stages; at t = 2min, ingress to a capillary, defined by an instant transition from a very low (CVF = $10^{-5}$) to a very high (CVF = 0.9) cell volume fraction together with opening of PIEZO1 channels for 0.4s; at t = 2min+0.4s, instant capillary egress simulated with a very low CVF ($10^{-5}$), PIEZO1 inactivation and medium composition restored to baseline values. The default period of the post-transit stage was set to extend to 5min. Within the DS1 baseline period, we set the data output conditions to be followed through DS1 and DS2: "accuracy" was set to 6, the minimal value found to be required for reporting the miniscule size of the predicted changes; "Epochs" was set to 111 and "Frequency factor" to $10^{-5}$, values found to offer acceptable point densities in the model outcomes for discerning predicted time-dependent trends unambiguously. For the DS2 period covering the brief capillary transit and

subsequent relaxation period back in the systemic circulation, we start by setting the overall duration of this stage (5min in the Ref sample) and then bring up the dedicated PIEZO routine confirming with "yes" the prompts on "Pz stage no or yes" and on "Restore Medium". All defaults are open to change as extensively implemented for Figs 3 and 5.

Comparison of results between simulations with CVF values of 0.9 and 0.00001 during single capillary transits showed substantial agreement, as anticipated because of the infinitesimal magnitude of the cell changes resulting from the brevity of the PIEZO1 open states. Nevertheless, the choice was made to retain the more realistic protocols with high-low CVF transitions in the presentation of the results, as differences of negligible size at a single transit level were shown to have significant effects when accumulated over the lifespan of the cells [1].

## Results and analysis

### Predicted volume effects of PIEZO1 activation during and after capillary transits

The four panels of Fig 3 show results abstracted from a large number of simulations, to illustrate the effects of the different permeabilities and permeability combinations used to represent the open state of the PIEZO1 channels. Fig 3A shows the effects of increasing PzCa at elevated PzA = 50h$^{-1}$. At PzCa = 0, RBC volume remains at baseline level. PzCa values > 0 elicit a biphasic response a sharp initial swelling followed by slow shrinkage towards below baseline levels. Increasing PzCa leads to higher initial peaks followed by faster and deeper dehydrations. In Fig 3B the Ref pattern with PzCa = 70h$^{-1}$ and PzA = 50h$^{-1}$ was followed for 60 minutes to expose the extremely slow short-term volume reversibility following post-transit dehydration. It takes about fourteen days for the quantal displacement to be fully restored back to baseline RCV levels, the result of the minute cation permeability of a RBC membrane with inactive channels, and of the diminutive displacements in the ion gradients driving net ion fluxes after biphasic responses.

Without a simultaneous increase in anion permeability (PzA = 0; Fig 3C, red), the swelling response to PzCa is markedly reduced thus exposing a powerful rate-limiting effect of the anion permeability on the peak response. Attribution of sodium and potassium permeabilities to PIEZO1, PzNa and PzK in Fig 3D (red), at the upper limit of measured values in on-cell patch-clamp experiments [31], caused only a minor parallel displacement upward on the Ref pattern. These results validate the exclusion of these permeabilities from the minimalist representation of the PIEZO1 effects in the chosen reference pattern, focussing all the attention on the two critical permeability components of the Ref response, PzCa and PzA.

The most important conclusion from the predictions in Fig 3 is that most of the PIEZO1--mediated effects on RBC volume during capillary transits follow from increased PzCa. PzA, PzNa and PzK modulate the response (Fig 3C and 3D), but without PzCa there is no response. PzA controls the magnitude of the peak response and is therefore the second most important component of the PIEZO1-mediated effects in the capillary context. This is particularly relevant to uncertainties concerning whether or not PIEZO1 is the unique mediator of both cation and anion permeabilities recorded in different experimental setups [31,48–50]. Obviously, model simulations cannot resolve this issue, but what they do show is that the coordinated activation of both permeabilities is critical for the optimized magnitude of the peak response, a suggestive argument for unique PIEZO1 mediation.

Three unexpected insights emerge from the predictions in Fig 3: Firstly, the infinitesimal magnitude of the PIEZO1-induced relative cell volume changes ($< 6^*10^{-5}$); secondly, their

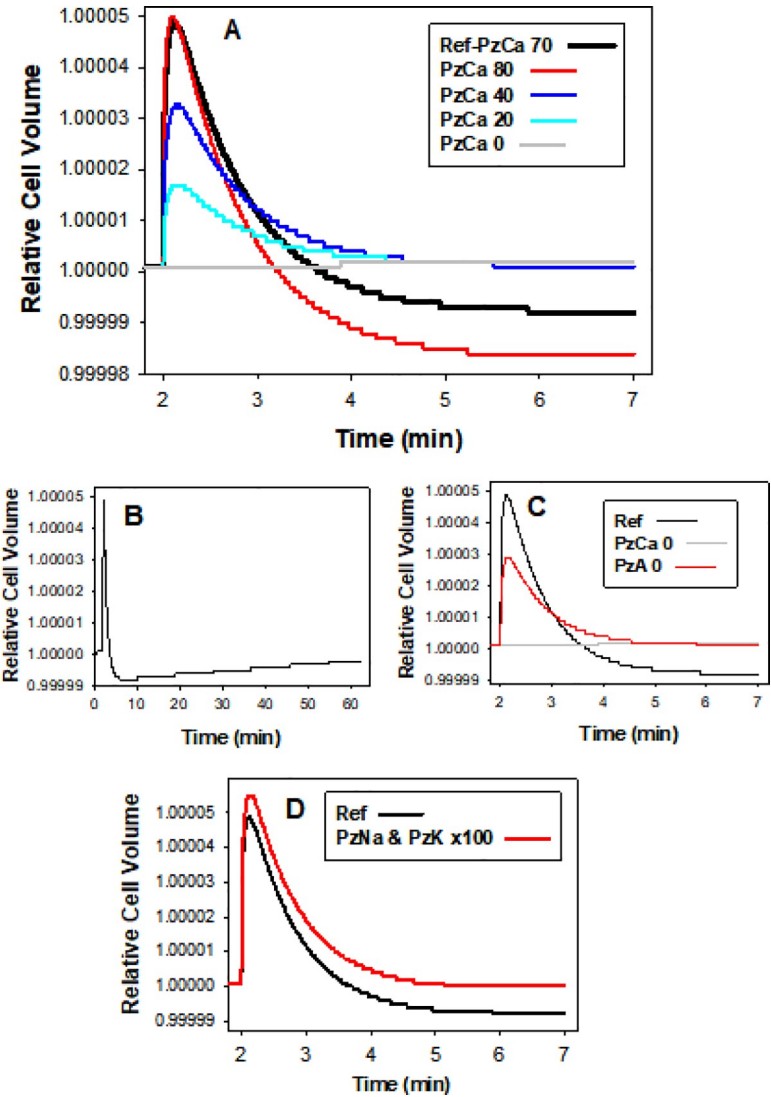

**Fig 3. Predicted volume effects of PIEZO1 activation during capillary transits.** After a baseline period of 2 min, ingress into capillaries at t = 2 min was simulated with a transition from a cell/medium volume ratio (CVF) of 0.00001 to 0.9 and a sudden opening of PIEZO1 channels activating electrodiffusional permeabilities to $Ca^{2+}$ (PzCa), monovalent anions (PzA; $A^-$ lumps $Cl^-$ and $HCO_3^-$), and monovalent cations (PzNa and PzK), as indicated. Capillary egress was set simultaneous with PIEZO1 channel closure (all PzX = 0) 0.4 seconds after ingress, together with a return to a CVF of 0.00001. RCV changes were followed thereafter for different lengths of time. Reference curves (Ref) are shown in black in all panels and were simulated with defaults of PzCa = $70h^{-1}$ and PzA = $50h^{-1}$. **A.** Effects of different PzCa values. **B.** Reference condition followed for 60min to estimate longer-term reversibility of PIEZO1-elicited volume changes. **C.** Short- and long-term effects of PzA = 0 relative to Ref with PzA = 50/h; PzA differences in the range 30 to 50/h, the range of values found to fit observed activated anion conductances under on-cell patch clamp [31], had no significant effects on Ref. **D.** Effects of increasing PzNa and PzK permeabilities to values 100-fold above those of the basal RBC $Na^+$ and $K^+$ permeabilities.

effective irreversibility within likely inter-transit times (Fig 2B), and thirdly, a kinetics dominated by sharp swelling during the narrow transits and by shrinkage after egress (Fig 3A), a seemingly paradoxical sequence for smooth flow through capillaries if the volume displacements are perceived as fulfilling a mechanical role, an impossible assignment for such miniature displacements.

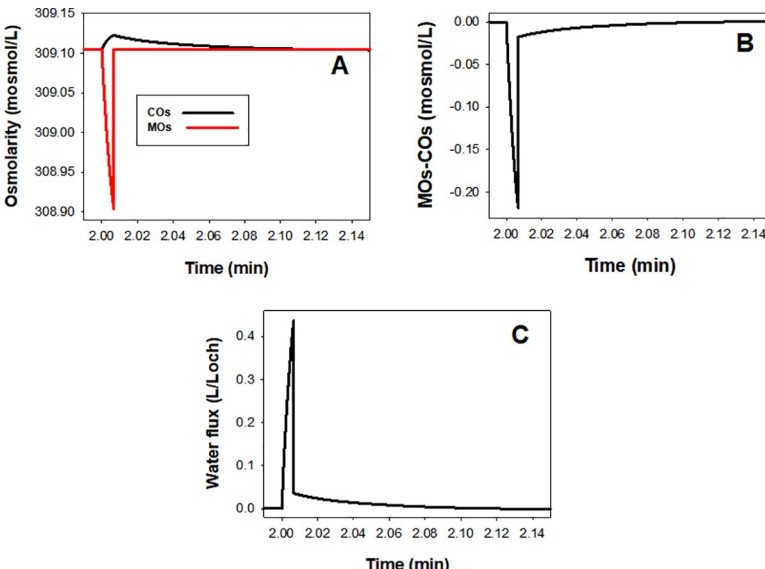

**Fig 4. Predicted changes in osmolarity (A), osmotic gradient (B) and water flux (C) across the RBC membrane during a capillary transit modelled following the reference protocol**. PIEZO1 activation triggers CaCl₂ influx increasing cell osmolarity (COs) during the 0.4s duration of the open state and capillary transit. With a cell/medium volume ratio of 0.9 during this period, the CaCl₂ shift from medium to cell caused a sharp fall in medium osmolarity (MOs) generating a gradient (B) for water influx (C) into the cells. After PIEZO1 closure and capillary exit, medium composition is instantly restored (vertical segments), simulating the return of the RBC to the systemic circulation, causing a sharp reduction in the osmotic gradient (B). Water influx persists for a few more seconds (C) shaping the peak response to PIEZO1 activation (Fig 2A), before all the downstream effects of CaCl₂ influx take over the slow dehydration phase of the biphasic response. Note the miniscule overall magnitude of the osmotic gradients and water fluxes on the expanded y-axis scales chosen to expose best their kinetic features.

## Mechanisms behind the predicted volume effects

We focus here on the reference condition (black curves) unless stated otherwise. PIEZO1 activation triggers CaCl₂ influx driven by the large inward electrochemical calcium gradient. CaCl₂ influx elevates cell osmolarity (COs) causing a sharp fall in medium osmolarity (MOs) (Fig 4A) thus generating a gradient for water influx into the cells (Fig 4B). After PIEZO1 closure and capillary exit, simulation of the instant return of the RBC to the systemic circulation is implemented by the CVF reduction and "Medium restore" routine (vertical segments in Fig 4). This causes a sharp reduction in the osmotic gradient, but not yet its disappearance. Although the water permeability of the RBC membrane is very high, the actual magnitude of the calcium-altered osmotic gradient, COs–MOs, is miniscule, hence water influx persists for a few more seconds (Fig 4C) shaping the peak of the volume response to PIEZO1 activation (Fig 3A) beyond the open state and before all the downstream effects of CaCl₂ influx take over the slow dehydration phase of the biphasic response.

The four main transport systems involved in the predicted volume response during capillary transits are PIEZO1, Gardos channels, PMCA, and the Jacobs-Stewart system (JS). The fluid flows contributed by each are indicated in the diagram of Fig 5 by the wide arrows W1, W2, W3 and W4, respectively, and further analysed in Fig 6. Fig 6A reports the effect of Gardos channel inhibition. It shows an elevated peak response followed by shrinkage towards the baseline level, never below. Any dehydration beyond baseline is therefore entirely the result of Gardos channel activity via W2 (Fig 5). Without Gardos channel activity, dehydration mediated solely by the PMCA (W3) and the JS (W4) mechanisms only accomplish restoration of the calcium, chloride and water initially gained via W1. The elevated peak response with

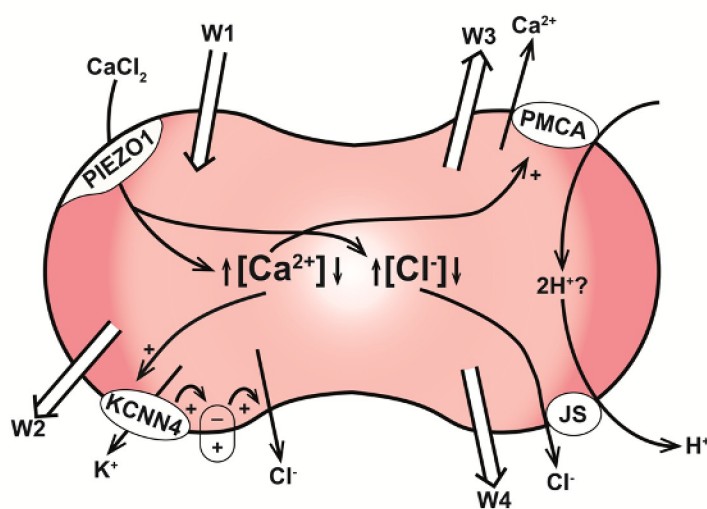

**Fig 5. Red blood cell volume control during capillary transits.** The figure illustrates the four main transporters involved in fluid dynamics throughout capillary transits in association with arrows (W) indicating the directions of net fluid transfers mediated by each: PIEZO1 (W1), KCNN4 (W2, Gardos channels), the PMCA (W3) and the Jacobs-Stewart mechanism operating like a $Cl^-$:$H^+$ electroneutral cotransport (W4, JS). PIEZO1 activation triggers $CaCl_2$ and fluid gain (W1) leading to a tiny but sharp increase in cell volume and $[Cl^-]_i$, and a much larger relative increase in $[Ca^{2+}]_i$. Elevated $[Ca^{2+}]_i$ activates Gardos channels (W2) and the PMCA (W3). Elevated $[Cl^-]$ stimulates HCl and fluid loss through the JS mechanism (W4). Gardos channel activation induces KCl and fluid loss (W2, Fig 5A). The PMCA, represented in the model as an electroneutral $Ca^{2+}$:$2H^+$ exchanger [60–62], extrudes the $Ca^{2+}$ gained via PIEZO1 concomitantly elevating $[H^+]_i$ which stimulates further HCl extrusion via JS (W4). The PMCA restores $[Ca^{2+}]_i$ to pre-transit levels within seconds of PIEZO1 closure (Fig 6C), setting the duration of the period during which dehydration is dominated by Gardos channel activity (W2, Fig 5D). The rate of JS-mediated dehydration (W4) depends on the proton and chloride concentration gradients and on the value attributed in the model to the JS turnover rate (Fig 6E), with a default Ref value based on early measurements [52]. The net water flux across the RBC membrane at each instant of time, Fw, can be expressed in this W representation by Fw = W1- (W2 + W3 +W4).

Gardos channel inhibited (Fig 6A, red relative to Ref) indicates that early Gardos channel activity (W2) also opposes slightly the peak-swelling led by W1.

## Effects of PMCA strength on the pattern of volume change

Fig 6, B to D report the predicted effects of different PMCA strengths on the PIEZO1 volume response (Fig 6B), on $[Ca^{2+}]_i$ levels (Fig 6C), and on calcium induced $K^+$ flux through Gardos channels (Fig 6D). The volume curves (Fig 6B) show that weak pumps generate the lowest peak heights and the fastest and more profound dehydration responses. This pattern is explained by the time-course of changes in $[Ca^{2+}]_i$ (Fig 6C) and of Gardos-mediated $K^+$ fluxes (Fig 6D). Fig 6C shows that a weak pump allows more calcium influx initially and takes longer than stronger pumps to extrude the calcium gained via PIEZO1. The three effects that result from enhanced Gardos channel activity in cells with weak calcium pumps are a larger initial Gardos-mediated $K^+$ efflux (Fig 6D), a reduced volume peak (Fig 6B), and a faster and deeper post-transit dehydration (Fig 6B).

According to Fig 6C and 6D, PMCA and Gardos channel activities are back to baseline within less than 20 seconds after PIEZO1 closure, even for the weakest pumps. This means that all dehydration mediated by the W2 and W3 components is completed during this period. Yet, dehydrations below baseline levels become apparent only minutes later, when the initial chloride gained during W1 has been fully restored to the medium via the JS cycle (W4). The

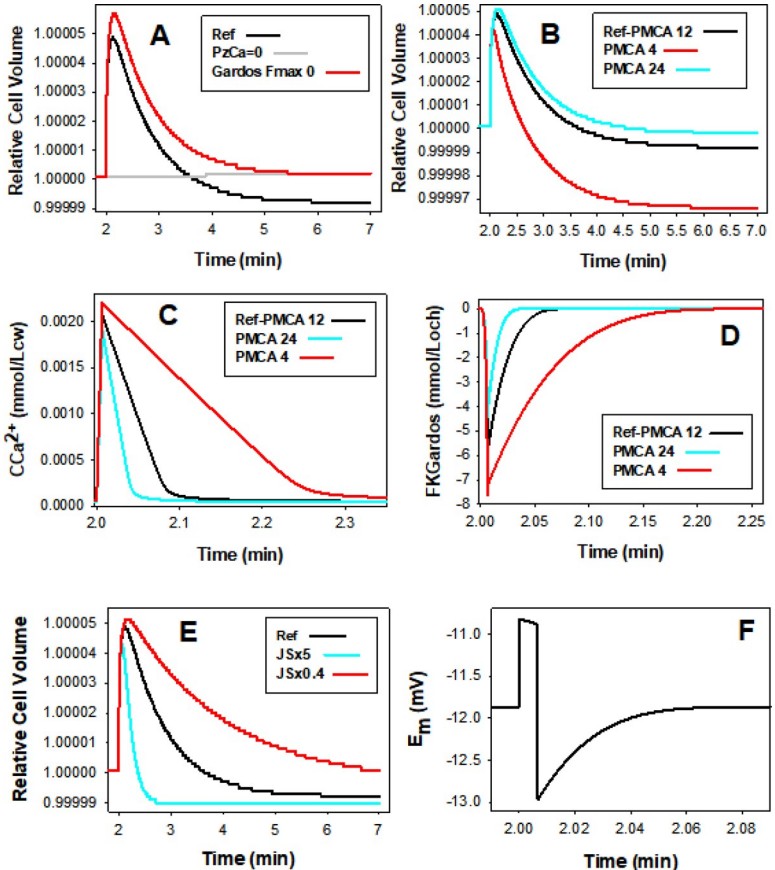

**Fig 6. Predicted effects of variations in the activities of Gardos channels, calcium pump and JS-mediated ion transport during capillary transits.** All effects reported relative to the default Ref conditions (black curves). Note the differences in x-axis time scales. **A.** Volume effects of setting FKmax though Gardos channels to 0 in Reference State. **B, C and D:** Effects of setting the PMCA FCamax values between 4 (cell with weak pump) and 24 mmol/Loch (cell with powerful pump) in the Reference State on RCV (**B**), on $[Ca^{2+}]_i$ (**C**, $CCa^{2+}$), and on potassium efflux through Gardos channels (**D**, FKGardos). **E.** Effects of reducing the default JS rate by 60% (JSx0.4) or of simulating it fivefold (JSx5) on the magnitude and rate of volume changes. **F.** Ref membrane potential changes during the ~5 seconds following PIEZO1 activation. Note that depolarization is sustained only during the 0.4s period of $CaCl_2$ influx through PIEZO1. After PIEZO1 closure the membrane potential immediately reverts to Gardos channel induced hyperpolarization, with a recovery kinetics set by $Ca^{2+}$ extrusion and Gardos channel deactivation.

comparative volume effects of changes in JS rate are shown in Fig 6E. It can be seen that the time it takes to expose Gardos-induced dehydration is determined by the displaced chloride concentration gradient and the JS transport rate (W4). These results show that the extent of dehydration elicited by Gardos channel induced KCl loss has to await the full extrusion of the PIEZO1 initial chloride load in order to be revealed (Fig 6B and 6E). The JS transport rate was found to vary among RBCs from adult healthy donors with a normal distribution and a coefficient of variation of 13–15% [51], variation to be noted for the likely spread of volume restoration rates around Ref (Fig 6E) for cells with different JS-AE1 transport rates.

## Influence of anion exchange (JS) and anion permeabilities (PzA and PA) on the kinetics of the volume response

JS flux dominates the kinetics of the late dehydration response (Fig 6E), but it also has a minor effect on peak height, suggesting that during the open state and the first few seconds after W1,

when W2 is the main hidden player, minor contributions from W3 and W4 are also able to influence the peak volume kinetics. It may seem puzzling that a JS rate with default values set at measured levels [52], capable of flux rates orders of magnitude larger than those of the PMCA and Gardos channels should mediate the slowest kinetic component of the PIEZO1 volume response. The answer rests with the negligible magnitude of the opposite displacements in the $Cl^-$ and $H^+$ driving gradients, which interested readers may wish to explore further using the detailed information in the csv data files, and the information on the JS phenomenology of the model explained in detail in the User Guide.

The rate at which Gardos-mediated KCl loss and dehydration (W2) proceeds during the few seconds after PIEZO1 closure is much reduced relative to the that during the brief open state period. This is because Ref PzA ($50h^{-1}$) is set about fifty fold higher than PA ($1.2h^{-1}$), the ground elecrodiffusional anion permeability of the RBC membrane [31,40,52–56]. Hence, after closure, Gardos-led dehydration becomes significantly more rate-limited by the anion permeability than during the brief open state. It is important to note, however, that most of W2-led dehydration takes place post-closure because of the extended period of elevated $[Ca^{2+}]_i$ (Fig 6C).

## Membrane potential changes during capillary transits

Fig 6F shows the membrane potential changes predicted for the first five seconds of the reference protocol following PIEZO1 activation. The initial depolarization during the $Ca^{2+}$ influx-dominated open state changes abruptly to Gardos-dominated hyperpolarization after closure, with a relatively slow return to baseline membrane potential following the calcium desaturation kinetics of the Gardos channels. The full magnitude of the predicted membrane potential displacement was about 2 mV, a miniaturized version of that fitting the results of the on-cell voltage clamp experiments with a much delayed PIEZO1 inactivation kinetics [31].

## Discussion

The results presented here predict the response expected of human RBCs during single capillary transits. The most important conclusion from these results is that despite the large variations in the magnitude of the responses caused mainly by differences in PIEZO1 and PMCA activities [23,57], the up-down biphasic volume pattern was always the same. The model predictions, solidly grounded on experimental data, provide a detailed high-resolution account of the changes in RBC homeostasis variables during and following capillary transits, changes triggered by deformation-induced activation of PIEZO1 channels on ingress of RBCs into capillaries. Analysis of the results and interpretation of the mechanisms involved was followed on large sets of simulations, summarily reported and analysed on Figs 3–6.

The main features of the predicted volume response of RBCs to a capillary transit initiated by PIEZO1 channel activation were i) a biphasic sequence of rapid initial swelling followed by slow shrinkage towards below-baseline volume levels (Fig 3A), ii) extremely slow post-transit reversibility (Fig 3B), and iii) the infinitesimal magnitude of the predicted volume displacements, incompatible with functional role assignments on capillary blood flow (Figs 3 and 6). The four main mechanisms involved in osmolite and fluid transfers during and following capillary transits are depicted in the diagram of Fig 5. PIEZO1 is responsible for initial swelling driven by $CaCl_2$ gain (Fig 3A). The PMCA restores calcium fully (Fig 6C); Gardos channel-mediated $K^+$ efflux causes a miniscule potassium depletion (Fig 6D), and the Jacobs-Stewart mechanism (Fig 6E) restores chloride and pH towards new levels when dehydration settles below baseline volume levels, strictly the result of early Gardos channel activity (Fig 6A). The

time-lag to the final dehydrated quasi-steady-state is largely determined by the chloride efflux rate through the Jacobs-Stewart mechanism (Figs 3B and 6E).

The main conclusions concerning the quantal hypothesis are that volume changes during single capillary passages have the potential to progressively accumulate, densify, swell or balance RBC volumes in the circulation depending on whether the duration of the inter-transit periods allows net dehydration, net swelling or balanced volume change to prevail (Fig 6B and 6E) during the changing homeostatic conditions of aging RBCs, alternatives to be investigated and tested in the companion paper [1].

The results reported here are not compatible with the $Ca^{2+}$-mediated adaptation mechanism suggested by Danielczok et al., [38]. According to this mechanism RBCs are supposed to experience relatively large and rapidly reversible volume reductions during capillary transits in aid of smooth capillary flow. This hypothesis was based on a perceived need for RBCs to be superdeformable on entering capillaries, for which a transient and sizeable initial volume decrease was considered necessary. GsMTx, a known inhibitor of PIEZO1 channels [58,59] was found to cause RBC shape changes, microfluidic block and reduced RBC filterability. Interpreting these effects as the result of PIEZO1 inhibition preventing the initial volume reduction necessary for transit, the GsMTx effects were considered supportive of the adaptation hypothesis. However, altered RBC filterability is not the normal condition of RBCs with normally silent PIEZO1 channels, suggesting that GsMTx or GsMTx-PIEZO1-cytoskelton interaction, besides channel block, induced additional abnormal side-effects in the conditions of their experiments, of doubtful relevance as tests of the adaptation hypothesis. In addition, as analysed in relation to Figs 3 and 6 here, the first volume deflection induced by PIEZO1 channel activation is swelling, not shrinkage. Above all, the known constitutive properties of human RBCs, as encoded in the model, do not allow volume changes of the magnitude, speed and reversibility required for the kind of flow dynamics suggested in the adaptation hypothesis.

## Supporting information

**S1 Fig. Accessing the red cell model (RCM).** The model programme and a comprehensive User Guide and Tutorial are available with full open access from a GitHub repository from the University of Glasgow (https://github.com/sdrogers/redcellmodeljava). This figure offers an introductory overview of the User Interface. **A: The Welcome Page.** The cartoon shows the main RBC components represented in model, including PIEZO1. At the bottom the user is offered two tags with alternative options to create a new simulated protocol, "New experiment" (on the left), or to transfer a previously saved protocol file "Load from file" (on the right). **B: The Central Page.** Contains two main panels and a set of six tags at the bottom. The left panel offers options for changing the initial (default) constitutive properties of the cell in the initial Reference State (RS). The right panel offers instructions for entering perturbations emulating experimental protocols and to implement the dynamic model responses. Protocol instructions are registered in sequential Dynamic State pages (DS). Bottom tags carry instructions for adding a DS stage, for running the model (Run), for saving a protocol file (Write protocol file), for seeking help with entries (Help), and for terminating the experiment (Close experiment). At the end of each run the user is offered options to plot any of the system variables for scrutiny or save the full output in a $^*$.csv file.
(DOCX)

**S1 Governing Equations. A full description of the equations governing the red cell model, including all relevant references.**
(PDF)

## Acknowledgments

The authors are grateful to Serge L. Y. Thomas and Daniel J. Lew for helpful comments and suggestions on the material contained in these papers and in the RCM User Guide.

## Author Contributions

**Conceptualization:** Virgilio L. Lew.

**Data curation:** Simon Rogers.

**Formal analysis:** Virgilio L. Lew.

**Investigation:** Virgilio L. Lew.

**Methodology:** Simon Rogers, Virgilio L. Lew.

**Software:** Simon Rogers, Virgilio L. Lew.

**Supervision:** Virgilio L. Lew.

**Visualization:** Simon Rogers.

**Writing – original draft:** Virgilio L. Lew.

**Writing – review & editing:** Simon Rogers, Virgilio L. Lew.

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
