## [Decision Letter · Decision Letter 0]

7 Oct 2020

Dear Dr Lew,

Thank you very much for submitting your manuscript "Up-down biphasic volume response of human red blood cells to PIEZO1 activation during capillary transits" for consideration at PLOS Computational Biology.

As with all papers reviewed by the journal, your manuscript was reviewed by members of the editorial board and by several independent reviewers. In light of the reviews (below this email), we would like to invite the resubmission of a significantly-revised version that takes into account the reviewers' comments.

We cannot make any decision about publication until we have seen the revised manuscript and your response to the reviewers' comments. Your revised manuscript is also likely to be sent to reviewers for further evaluation.

Sincerely,

Daniel A Beard

Deputy Editor

PLOS Computational Biology

Daniel Beard

Deputy Editor

PLOS Computational Biology

Reviewer's Responses to Questions

**Comments to the Authors:**

Reviewer #1: The manuscript proposed a very interesting question about how red cell volume changed in response to Piezo1 activation during capillary transit. In this case, computational approach was used to simulate red cell volume upon piezo1 activation and demonstrated that cell volume increased following by shrinkage during capillary transit and the magnitude of such volume change was small.

Comments

1) The kinetic of such process is not clear. For example, start from time 0 where red cells enter capillary, how long it takes to activate piezo1 channel and what is the timescale in other related signaling pathways that eventually lead to cell volume change (figure 4)? This kinetic analysis is critical to understand the question proposed by the authors.

2) It is not clear to me how the magnitude of volume change was calculated. Doe it based on the number of piezo1 channels on the cell membrane?

In terms of the second manuscript entitled “PIEZO1 and the mechanism of the long circulatory longevity of human red blood cells”, the description was comprehensive but it would be better if it could be more focused. For example, it would be interesting to focus on quantitatively how decay in different channel activities that regulated cell volume contribute to red cell longevity and what were the relative roles of each channel in this process.

Reviewer #2: Reviewer’s report

In this study, Rogers and Lew present a new computational model of the human red cell (RCM) aiming at quantifying transient states of cell water and -electrolyte exchanges during capillary transits. The duration of a single transit spans from 0.5 to 1.5 s, where the cells transiently squeeze and deform before reemerging from the capillary into the main circulation for reestablishing homeostatic parameter values. The general problem dealt with concerns the widely accepted presumption (indirectly experimentally investigated) that the change in composition and volume of single RBCs during a single capillary transit is not fully reversible for which the term ‘quantal change’ has been introduced. Following the entering of cells into a capillary (denoted ingress), the cell/medium volume ratio (CVF) changes from 0.00001 to 0.9, that is, the ratio approaches one during transit. The nature of the problem implies that during a single capillary transit the two-compartments of cells and the medium surrounding the cells, respectively, are submitted to non-trivial time dependent states.

The RCM is designed for computing stationary and nonstationary states of red cell volume and density, ion compositions (Na+, K+, Cl-, Mg2+, and Ca2+), pH, and membrane potential. The mathematical treatment is presented in ‘Governing Equations of the Red Cell Model (RCM)’ of two sections, The initial Reference State (RS), and The dynamic state (DS), respectively. The RS describes the initial condition of the system prior to capillary ingress. The conditions defining the steady state are governed by impressive details provided by previous experimental studies contributed very significantly by Lew and his associates in the UK and abroad. Thereby the structure of equations and ranges of free parameter values for computing homeostatic variables are given with good precision. The main systems of equations carrying transmembrane fluxes are the mechanosensitive, non-selective PIEZO1 channels, the Ca2+ sensitive Gardos channels, the PMCA and the Jacobs-Stewart mechanism. Flux equations, all of conventional structure, buffer expressions handling cytosolic pH, Mg2+ and Ca2+, and the main four transport systems involved together with expressions of electroneutrality, osmotic equilibrium, and mass conservation are solved by an iterative Newton-Raphson’s method.

An instant CVF transition from 0.00001 to 0.9 and the simultaneous activation of the non-selective PIEZO1 channel simulate ingress of a red cell in a capillary for initiating the DS. Similarly, after 0.4s egress is simulated by an instant PIEZO1 closure and return of CVF to 0.00001 corresponding to the return of the red cell to the systemic circulation of a plasma-like composition.

Computations given by the model indicate how the brief PIEZO1 activation initiates a sharp ‘swelling peak’ followed by a much slower recovery period towards a lower-than-baseline volume. Further inspection revealed that the cell-volume on-response was due to a net CaCl2 and fluid gain via the PIEZO1 channel driven by the inwardly directed electrochemical Ca2+ gradient. The subsequent cell volume loss followed a complex time-course with sequential, but partially overlapping contributions by KCl loss via Ca2+-activated Gardos channels, restorative Ca2+ extrusion by the plasma membrane calcium pump, and chloride efflux by the Jacobs-Steward mechanism. The key finding was that the change in relative cell volume during a single capillary transit is not fully reversible but attains a value around 10-5. Obviously, this infinitesimal volume change is of no functional role in capillary flow physiology.

Evidently, however, the above biphasic cell volume response predicted by the RCM would conform to the quantal hypothesis. In a follow-up computational study, the authors intend to investigate the obvious question whether the cumulative effects of subsequent quantal responses would account for the observed changes in density in the cause of RBC senescence.

The present work is elegantly conceived, methodologically sharpened, and convincingly carried out. All of the mathematical expressions defining ion- and volume fluxes across the red cell membrane are of a reversible nature. Therefore, my immediate critique of this overall impressive study is that I miss an explicit discussion of whether the quantal effect simply is due to a too brief inter-transit period. If exit from the post-transit observation period were extended to infinity, according to my understanding the quantal response would be eliminated. If this does not hold true, I am not too well informed by the authors’ well written manuscript.

Reviewer #3: General comments:

The authors stated goal is to address a fundamental question in human physiology – what sets the lifespan of RBCs? Their stated plan is extremely ambitious, seeking to model events occurring on timescales ranging from < 1s to 100 days. The authors’ combined work in this manuscript, supplement, online repository, and companion paper represents a tremendous effort and a rich resource for the community that should be shared in some forum.

The results reported in the main body of this manuscript rely on the validity of the RCM model. Model code is available online, and a summary is provided in the appendix. It does not appear that the model itself has been fully peer-reviewed overall. The appendix provides many details on calculations of fluxes, concentrations, etc., but significant additional detail and motivation would be needed for a complete review, and it seems that this model needs to be reviewed in a manuscript of its own.

This manuscript reports a study of one membrane channel under one set of narrowly defined conditions and one set of assumptions. These limitations and this narrow focus need to be explicitly stated. The manuscript in its current form suggests that the findings are far more broadly applicable. The main reported finding of this paper is a net relative volume loss of 10e-5 per RBC after a 0.5s – 1.5s capillary transit under the assumptions and simplifications listed. No confidence interval is provided, and no uncertainties are provided with the model. It’s very hard to believe that this result can be distinguished from zero with confidence.

The authors make many simplifying assumptions, which are reasonable but must be acknowledged and clearly labeled as assumptions, and the scope of the study must be appropriately narrowed. As an example of one of these many assumptions, the authors restrict capillary transit times to 0.5-1.5s. There are obviously capillary transit times in vivo well outside this range, not to mention reports of pulsatility in at least some capillaries (https://elifesciences.org/articles/45077), which is not studied. The authors also do not allow for any mechanical changes in the material of the RBC membrane. If they require their model to have precision needed to detect relative changes at the level of 10e-5, these and other complexities need at least to be discussed. At relative volume changes this small, the compressibility of water may even become a factor.

Specific comments:

Response to Editor: In the “Response to Editor,” the authors write that they used “the only experimental result in the literature approximating the dynamic in vivo events during capillary transits.” Without a definition of “dynamic in vivo events” and “approximating,” it’s difficult to assess that statement. On the one hand, there are many in vivo studies which do more than just approximate in vivo events. On the other hand, there is vast literature on in vitro studies of blood flow, and it’s not clear where these other studies are deficient in the authors’ view.

Abstract: The abstract should make clear right away what variables are being studied. The title makes clear that the authors are simulating the effects of the PIEZO1 channel on RBC volume, but the first sentence of the abstract mentions only “homeostasis” which includes far more than just volume.

Introduction: The primer is thoughtful and helpful but includes some oversimplifications that may mislead readers. For instance, RBC volume is claimed to be maintained “within a narrow margin around 60%” of a sphere with the same diameter, but other studies show a different range of S/V ratios: (https://pubmed.ncbi.nlm.nih.gov/27354532/).

Introduction: The authors report that “RBCs undergo gradual densification for most of their lifespan.” They need to be specific about what they mean by “gradual” and “most of their lifespan” because there is conflicting and contradictory literature on this subject. Reference #12 in their paper states “most of the density change occurs early in the RBC lifespan.” This statement would contradict many people’s interpretation of the author’s description.

Introduction: The authors’ discussion of RBC turnover is oversimplified: “The terminal clock is activated by immune signaling…” That’s one of many proposed mechanisms, all with evidence, but none shown to be necessary. See helpful commentary very recently in Blood: “Although multiple cellular and biochemical changes that correlate with RBC age have been documented, the specific changes that lead to clearance of senescent RBCs from circulation remain unclear.” (https://ashpublications.org/blood/article/136/14/1569/463973/Turning-over-a-new-leaf-on-turning-over-RBCs).

Methods: The authors claim that ref. 36 reports “the closest experimental approximation to capillary flow conditions available to date.” This statement needs to be justified. Given the large number and variety of in vitro and in vivo studies of the microcirculation, it’s hard to believe. Furthermore, if the authors are studying shear stress, they need to consider arterioles where the shear rates are reportedly highest in vivo.

Methods: The authors say that a “full explanation for the value of 70h-1 … will have to await … the companion paper.” The readers need some more clues here.

Results: The “Mechanisms behind the predicted volume effects” paragraph appears to provide a concise description of the scope and findings of this simulation study. The scope of the presentation of the overall manuscript might be better off if narrowed similarly, assuming the RCM model’s validity is also established.

Figure 1: the fit to the data is very poor. I am not convinced that this data set provide any meaningful constraint on the space of potential models. This figure also appears to contain the only validation of any simulation results in the paper. All other figures appear to show only simulated data and no experimental validation results.

Reviewer #4: This is a quantitative and rigorous analysis by the world's leading expert on red blood cell physiology of the time dependence of the ionic composition and volume of red cells subjected to perturbations, such as calcium ion influx from red cell distortion in the capillaries. I have nothing but praise for this work, which demonstrates a comprehensive understanding of experiments and all theoretical aspects of the subject. Both articles are written with great clarity, so, although quite technical, they can be easily understood by any non-specialist who is willing to take the time to read them carefully. The articles are also quite timely, as there is a growing interest by hematologists in understanding the lifetime of red cells cell and its role in understanding disease pathogenesis and gene therapy for inherited disorders such as sickle cell anemia.

**Have all data underlying the figures and results presented in the manuscript been provided?**

Reviewer #1: None

Reviewer #2: Yes

Reviewer #3: Yes

Reviewer #4: Yes

PLOS authors have the option to publish the peer review history of their article (what does this mean?). If published, this will include your full peer review and any attached files.

Reviewer #1: No

Reviewer #2: No

Reviewer #3: No

Reviewer #4: No
---

## [Decision Letter · Decision Letter 1]

17 Nov 2020

Dear Dr Lew,

Thank you very much for submitting your manuscript "Up-down biphasic volume response of human red blood cells to PIEZO1 activation during capillary transits" for consideration at PLOS Computational Biology.

As with all papers reviewed by the journal, your manuscript was reviewed by members of the editorial board and by several independent reviewers. In light of the reviews (below this email), we would like to invite the resubmission of a significantly-revised version that takes into account the reviewers' comments.

Although most reviewers are satisfied with and recommend publication of both of your companion papers, Reviewer 3 raises a number of important issues that ought to be addressed. In fact, I think that those concerns are addressed in Paper #1, which is focused on development and identification of the model. I think that I am partly to blame for not effectively communicating to reviewers that this paper is part of a set, and making sure that reviewers knew how to find both papers.

I request to provide a clear point-by-point response to all of the issues raised by Reviewer 3, some of which (but not all) are already be addressed in the companion paper. We cannot make any decision about publication until we have seen the revised manuscript and your response to the reviewers' comments. Your revised manuscript is also likely to be sent to reviewers for further evaluation.

Sincerely,

Daniel A Beard

Deputy Editor

PLOS Computational Biology

Daniel Beard

Deputy Editor

PLOS Computational Biology

Reviewer's Responses to Questions

**Comments to the Authors:**

Reviewer #1: The manuscript is improved after revision. There is one article i think it might help when the authors discussed kinetic issues: Cinar et al. 2015, PNAS, "Piezo1 regulates mechanotransductive release of ATP

from human RBCs"

Reviewer #2: None

Reviewer #3: In this manuscript in revised form, the authors report simulation results from a novel java implementation of equations describing kinetics of some in vivo RBC biological processes and the use of these simulation results to infer quantitative details of in vivo RBC volume changes during circulation. Three questions need to be answered to assess the authors’ claims regarding their claim of infinitesimal volume changes in respond to in vivo shear stress: (1) Does their java code correctly implement the model equations? (2) Are their model equations accurate descriptions of the biological processes they intend to represent? (3) Do those equations collectively describe enough biological processes to enable accurate prediction of in vivo changes in RBC volume during circulation?

For (1), the code review required is beyond the scope of this manuscript review. For (2), the authors claim that their equations have been peer reviewed and published in papers starting at least as early as 1986. For (3), readers need volume measurements to validate simulation results. None are provided. The poor fit to experimental data in figure 1 is explained in the authors’ response as something other than experimental validation of their model. It seems that there is no attempt to validate their model of the effect of shear stress on RBC volume with experimental data. Given many modern methods for measuring RBC volume, this model and its surprising claim must be validated.

As noted in the first review, the authors make many simplifying assumptions and limit their focus to a narrow range of conditions, for instance only considering events that occur in capillaries, ignoring mechanical effects, cellular material properties, and different conditions present in vivo, for instance, in arterioles where shear rates are much higher, or pulsatile flow. They write in their reply that a justification for the narrow focus is provided in the “Preliminary RCM tests” section, but there is no mention of mechanical forces, other vessels, or pulsatility. It is an enormous stretch to ask readers to believe that a model devoid of mechanics can accurately predict a volume change on the order of 10e-5.

The first review pointed out several misleading oversimplifications in the “primer” section, including overly narrow claims of surface-to-volume ratio, claims about gradual densification that contradict references cited, and skewed summary of mechanisms of RBC turnover. They revise their statement on RBC turnover but still only mention one mechanism. For the other oversimplifications, the authors reply that these details are not needed in a primer, but the primer is part of a scientific paper where accuracy and balanced reports of current knowledge are essential.

The authors were asked to justify their statement that Danielczok et al. reports “the closest experimental approximation to capillary flow conditions available to date.” They explain that Danielczok studies PIEZO1 in a unique way, but their revised manuscript does not clarify this point and still justifies the sole focus on Danielczok based on “capillary flow conditions.” They also explain that they are not using the Danielczok data for validation, which leaves no experimental validation.

Reviewer #4: Valuable and outstanding work!

**Have all data underlying the figures and results presented in the manuscript been provided?**

Reviewer #1: None

Reviewer #2: Yes

Reviewer #3: Yes

Reviewer #4: Yes

PLOS authors have the option to publish the peer review history of their article (what does this mean?). If published, this will include your full peer review and any attached files.

Reviewer #1: No

Reviewer #2: No

Reviewer #3: No

Reviewer #4: No
---

## [Editor Report · Decision Letter 2]

14 Jan 2021

Dear Dr Lew,

We are pleased to inform you that your manuscript 'Up-down biphasic volume response of human red blood cells to PIEZO1 activation during capillary transits' has been provisionally accepted for publication in PLOS Computational Biology.

Best regards,

Daniel A Beard

Deputy Editor

PLOS Computational Biology

Daniel Beard

Deputy Editor

PLOS Computational Biology

---

## [Editor Report · Acceptance letter]

13 Feb 2021

PCOMPBIOL-D-20-01635R2 

Up-down biphasic volume response of human red blood cells to PIEZO1 activation during capillary transits

Dear Dr Lew,

I am pleased to inform you that your manuscript has been formally accepted for publication in PLOS Computational Biology. Your manuscript is now with our production department and you will be notified of the publication date in due course.

With kind regards,

Alice Ellingham
